# Non-Destructive Impedance Monitoring of Bacterial Metabolic Activity towards Continuous Lead Biorecovery

**DOI:** 10.3390/s22187045

**Published:** 2022-09-17

**Authors:** George Andrews, Olga Neveling, Dirk Johannes De Beer, Evans M. N. Chirwa, Hendrik G. Brink, Trudi-Heleen Joubert

**Affiliations:** 1Carl and Emily Fuchs Institute for Microelectronics (CEFIM), University of Pretoria, Pretoria 0002, South Africa; 2Department of Chemical Engineering, Faculty of Engineering, Built Environment and Information Technology, University of Pretoria, Pretoria 0002, South Africa

**Keywords:** nondestructive, inline monitoring, bacterial growth, metabolic activity, lead biorecovery, impedance spectroscopy

## Abstract

The adverse health effects of the presence of lead in wastewater streams are well documented, with conventional methods of lead recovery and removal suffering from disadvantages such as high energy costs, the production of toxic sludge, and low lead selectivity. *Klebsiella pneumoniae* and *Paraclostridium bifermentans* have been identified as potential lead-precipitating species for use in a lead recovery bioreactor. Electrical impedance spectroscopy (EIS) on a low-cost device is used to determine the potential for the probe-free and label-free monitoring of cell growth in a bioreactor containing these bacteria. A complex polynomial is fit for several reactive equivalent circuit components. A direct correlation is found between the extracted supercapacitance and the plated colony-forming unit count during the exponential growth phase, and a qualitative correlation is found between all elements of the measured reactance outside the exponential growth phase. Strong evidence is found that Pb(II) ions act as an anaerobic respiration co-substrate for both cells observed, with changes in plated count qualitatively mirrored in the Pb(II) concentration. Guidance is given on the implementation of EIS devices for continuous impedance monitoring.

## 1. Introduction

The presence of lead in wastewater poses a major concern due to health risks including neurodevelopmental alterations and neurodegeneraton, disruption of cellular metabolisms through bonding with essential molecules including hemoglobin synthesis, and the occurrence of anaemia, diarrhoea, stomach ache, and kidney damage [1,2,3]. The main sources of lead pollution are anthropogenic, most prominently the production of lead acid batteries, fertilizers, and pesticides, as well as from industrial wastewater effluents and mining waste [4]. It is estimated that environmental Pb concentrations have increased by three orders of magnitude in the last 300 years due to these anthropogenic activities [5]. In addition to the environmental and human health risks posed by Pb, the world lead reserves are rapidly declining, with the world mining of raw lead estimated at 4.3 Mt and an estimated remaining global Pb reserve of 90 Mt, meaning that *circa* 20 years’ supply of Pb reserves remains [6].

Various conventional methods of lead recovery and removal are currently in operation, such as ion exchange, electrowinning, electrocoagulation, cementation, and reverse osmosis. The disadvantages of these methods include the high energy costs, the production of toxic sludge, and the low selectivity for Pb [7]. Consequently the bioremediation of Pb is an attractive alternative method for lead removal and recovery. This is due to its relatively low cost and high selectivity to Pb, resulting from lead-specific resistance mechanisms, including efflux mechanisms, changes in cell morphology, extracellular sequestration, intracellular bioaccumulation, siderophore production, biosorption, and bioprecipitation [5].

Pb(II) bioprecipitation by an industrially obtained microbial consortium has been previously investigated by the research team [8]. The consortium was obtained from a battery recycling plant in South Africa, with the dominant Pb-precipitating species identified as *Paraclostridium bifermentans* (a Gram-negative bacterium, rod-shaped and measuring 2 µm by 0.5 µm) and *Klebsiella pneumoniae* (a Gram-positive spore-forming bacterium, rod-shaped and measuring 2 µm by 0.5 µm). It was demonstrated that the precipitate included elemental Pb, evidencing an anaerobic respiration mechanism with Pb(II) as the terminal electron acceptor/co-substrate [8,9]. The bioreduction of Pb(II) to elemental Pb is of industrial importance as it provides a potential route for the valorization of Pb(II) [8]. The same consortium displayed continuous removal efficiencies in an upflow anaerobic sludge blanket (UASB) reactor of between 90 and 100% for inlet Pb(II) concentrations varying between 80 and 2000 mg/L Pb(II). A maximum removal rate of 1948.4 mg/L/d was achieved, while the presence of elemental Pb in the precipitate was confirmed [10].

Metabolic activity has been monitored with non-invasive optical methods such as fluorescence [11] or dark-field microscopy [12], but these are usually expensive laboratory-based methods that require trained operators and are challenging to miniaturize and transfer to the field. Electrochemical techniques offer a non-destructive approach to monitor microbial activity and growth and provide potentially low-cost field tools for monitoring the status of bioprocesses. Considerable electrochemical research has focused on the use of electrodes for electricity production in microfuel cells, as well as for monitoring the anchored electroactive biofilms in the same context. Application of electrochemical methods for contaminant conversion and recovery is an emerging field.

In the context of the lead recovery application, studies report that microbes improve their catabolism by using conductive minerals as conduits of electrons [13]. Microorganisms grown in anoxic conditions may act as electron acceptors/donors and biocatalytically utilize organic and inorganic substrates in the bioreactor media as electron donors/acceptors. Electron transfer typically occurs via one or more of three pathways: (i) direct electron transfer through the outer cell membrane, (ii) electron shuttle molecules mediated by cells, and (iii) conductive nanowire protein cell projections [14]. During electrochemical monitoring, the electrode also acts as an electron donor/acceptor for the microorganisms. According to the literature, both the organisms in the current study are electrochemically active. *P. bifermentans* interacts with insoluble electron acceptors or donors through extracellular electron transfer [15], whereas *K. pneumoniae* (strain L17) transfers electrons between the cells and the anode electrode in a microfuel cell via a recyclable electron shuttle molecule [14].

The most commonly used electrochemical method for microbial metabolic analysis is cyclic voltammetry (CV), which provides a measure of the reduction/oxidation (redox) activity and the electron transfer phenomena of electroactive bacteria. CV is performed with a potentiostat instrument in an electrochemical cell with a three-electrode transducer (and sometimes a four-electrode transducer). The working principle for CV is to apply a linear ramp potential at a specific scan rate and measure the resulting current to determine the reduction and/or oxidation potentials of the electrochemical system. In many CV cases, a redox probe is added to the system that often requires sample pre-treatment [16]. A recent example for the CV detection of *K. pneumoniae* spiked in 0.01 M phosphate-buffered saline (PBS) contained 5 mM K_4_Fe(CN)_6_/K_3_Fe(CN)_6_ in a ratio of 1:1 as a redox couple [17]. CV measurements do not necessarily require reagents that act as labels for the bacterial cells under investigation, but the method may benefit from the use of labels to amplify the electrochemical response, as seen in the microbial fuel cell exploration for an Escherischia coli sludge [18]. Label-free examples of *K. pneumoniae* CV-derived detection in the context of biofilms require complex electrodes to immobilize cells [19,20].

Electrochemical impedance spectroscopy (EIS) is used to understand many phenomena in electrochemical systems and is currently receiving research attention for its versatile and facile field analysis. EIS is performed in an electrochemical cell with a two-electrode transducer, most often an interdigitated electrode (IDE) structure, and requires minimal sample preparation [21]. The working principle for EIS is to apply a small-amplitude sinusoidal potential across a range of frequencies and measure the resulting current to determine both the magnitude and phase of the frequency-dependent electrical impedance. The advantage of a large frequency range is that different phenomena provide information on the time constants of the physical processes within the system that affect the electrical impedance of a sample at different frequencies. From a few Hz to approximately 100 kHz, the impedance is typically dominated by the double-layer capacitance at the electrode–solution interface [22]. At higher frequencies up to low MHz, the impedance is usually determined by the solution conductivity, the cell membrane potential, and the displacement of charged ions surrounding the charged cell membrane. For frequencies above 1 MHz, the impedance is associated with intracellular structures such as the dielectric lipid membranes and conductive cytoplasm and periplasm, whereas at very high frequencies above 20 MHz, molecular polarization and relaxation can become significant [21,23]. At frequencies lower than a few Hz down to mHz, time constants associated with solid-phase particle diffusion processes in the electroactive material may be measured [24].

An important property of EIS is that parameters for an equivalent electrical circuit are extracted from the measured impedance data via regression analysis [25], providing not only better interpretation of physicochemical processes, but also enabling the use of computer simulation to evaluate specific growth properties [21]. A further advantage of modeling the cells with polarizable circuit elements is that EIS is not limited to microorganisms demonstrating electroactivity [25]. The most prominent equivalent circuit parameters used for impedance-based metabolic activity monitoring in the literature are: (i) the conductivity of ohmic media as used in the measurement of the ionic resistance [26], (ii) the polarization resistance associated with the charge transfer of reduction and/or oxidation mechanisms on the electrode surface, as used in the measurement of biofilm thickness [26], and (iii) a constant-phase element representing the double-layer capacitance due to charge build-up at the boundary between a typical electrode surface and the solution, as discussed for several aqueous solutions using planar electrodes [27], and for the adhesion monitoring of a *Pseudomonas aeruginosa* PA14 wild-type and wspF mutant bacterial biofilm without a redox probe [28]. The Randles equivalent circuit model is often used for the fitting of multiple impedimetric parameters simultaneously [29]. As one example, an interesting machine learning algorithm for fitting both resistance (related to charge transfer) and capacitance (associated with the electrode surface double layer) improved accuracy in determining bacterial *E. coli* concentrations [30].

An in-situ batch assay investigation of *Clostridium phytofermentans* included both CV and EIS [31] and provides an excellent comparison of these techniques. The CV showed a reduction peak in the presence of bacteria, with a good correlation seen between peak measured current and electron transfer at the surface of the cells. It was confirmed that the increased charge transfer resistance was a function of the carbon and energy source depletion of the cells, and not associated with the bulk growth medium. EIS was performed between 0.1 Hz and 100 kHz at a 50 mV amplitude. The equivalent circuit model consists of the series combination of (i) a resistor representing the electrolyte resistance, and a parallel connection consisting of (ii) a parallel combination of a resistor and a constant-phase element for low to medium frequencies, and (iii) a series combination of a resistor and a constant-phase element for high frequencies. Changes in the medium-frequency resistance are ascribed to ionic flux across the bacterial membrane, whereas variation in the high-frequency resistance is ascribed to metabolic activity and growth. The other circuit parameters did not change much. The extraction of the charge transfer resistance indicated the decrease in growth rate as the carbon and energy source diminished. Using the additional information provided by EIS allows for the close correlation of a strong ionic flux during culture germination before metabolic carbon source utilization. EIS data provided a broader view of the system electrochemistry.

The current research investigates the feasibility of the impedance monitoring of the bacterial metabolic activity of the active Pb-precipitating species (*P. bifermentans* and *K. pneumoniae*) that were isolated from the lead-resistant battery recycling plant consortium [8]. Initial studies must be performed containing a single microbial strain to provide an understanding of the measured electrochemical response due to the reactions at the electrodes in the complex system. To measure the impedance of the biological cells in the complex medium, a custom low-cost impedance analyzer [32] is used without the addition of probes or labels. Although they are used often, typical standard equivalent circuits do not fit experimental bacterial data well [22,29], and a complex non-linear squares regression fit to a polynomial associated with different reactive components is proposed. There is an association between the rate of substrate metabolism and the microbial concentration [13], and therefore the experimental impedance data can be related to gold-standard analytical measurements of the microbial concentration and the residual Pb(II) substrate ion concentration of the same samples.

## 2. Materials and Methods

### 2.1. Cell Sample Preparation

#### 2.1.1. Preparation of Pb(NO_3_)_2_

The Pb(II) solution was prepared by adding 1.6 g of Pb(NO_3_)_2_ (Glassworld, South Africa) to 100 mL distilled water. This produces a stock solution of 10,000 mg/L Pb(II).

#### 2.1.2. Batch Reactor Preparation

The batch reactors were prepared using 20 g/L tryptone (Oxoid, South Africa), 10 g/L yeast extract (Oxoid, South Africa), and 1 g/L NaCl (Glassworld, South Africa) in distilled water. The broth solution was placed in 100 mL serum bottles with 0.8 mL of the 10,000 mg/L Pb(II) stock solution added if the experiment required Pb(II), resulting in a solution with an 80 mg/L initial Pb(II) concentration. The reactors were inoculated with 0.2 mL preculture stored with 20% glycerol. After inoculation, the reactors were purged for approximately 3 min with N_2_ gas to ensure anaerobic conditions, and then sealed. The reactors were continuously incubated at 35 °C during sampling. Samples were taken using a sterile hypodermic needle and syringe and stored in sterile 2 mL vials. The samples were taken during the expected exponential growth phase and the expected stationary phase.

### 2.2. Colony-Forming Unit Determination

The colony forming units (CFU) of each experiment were determined using the spread plate method on agar. For the reactors not containing Pb(II), the agar consisted of 10 g/L tryptone (Oxoid, South Africa), 5 g/L yeast extract (Oxoid, South Africa), 10 g/L NaCl (Glassworld, South Africa), and 15 g/L agar in distilled water. For the reactors containing Pb(II), the agar consisted of 20 g/L tryptone (Oxoid, South Africa), 10 g/L yeast extract (Oxoid, South Africa), and 1 g/L NaCl (Glassworld, South Africa) in distilled water. The sample was diluted to a range of 10^−4^ to 10^−7^ in sterile distilled water. The number of colonies was counted 96 h after plating was completed.

### 2.3. Residual Pb(II) Measurements

The residual Pb(II) concentration was determined by centrifuging the samples at 9000 RPM for 10 min (Hettich, Universal 320 R, Tuttlingen, Germany). The supernatant, which contained Pb(II) that was not precipitated, was diluted in distilled water and the concentration of Pb(II) was measured using an atomic absorption spectrometer with a Pb lumina hollow cathode lamp (PerkinElmer AAnalyst 400, Waltham, MA, USA). The dilution was done to ensure that the concentration of Pb(II) lay within the limits of the analytical instrumentation.

### 2.4. Electrical Measurement Setup

Hourly samples of 1.8 mL were placed in cuvettes to monitor the growth of cells via electrical impedance measurements. A Metrohm Dropsens boxed connector DSC4MM was placed in a custom 3D-printed jig that held the electrodes in the same position within the bulk liquid at the top of the cuvette. The Dropsens PW-IDEAU100 screen-printed electrodes used for EIS experiments had 100 μm track width gold interdigitated electrodes (IDEs) on a white plastic substrate [33]. Figure 1 shows the measurement jig used.

Using IDEs in impedance spectroscopy techniques enables measurement of the electrical properties of the volume close to the electrode surface [22]. The distance of sensitivity from the surface of the electrode is known to be related to the spacing of the interdigitated tracks [26], so the given electrode is sensitive for 100 μm from the surface. This provides an acceptable volume of culture to be interrogated, despite the fact that the sensitivity might be improved if the electrode feature dimensions are approximately the same as the microbe size [26]. Although a wide variety of electrode geometries and sizes are commercially available, this electrode is selected over more complex geometries or smaller feature sizes because the research group standardizes on electrode configurations that can be fabricated using low-cost additive manufacturing technologies such as screen printing and inkjet printing.

Beyond the type of bacterium, the electrochemical current output depends only on the type of utilized substrate and the interface materials on the electrode [13]. The conductive matrix for the microbes in the lead biorecovery application is expected to enhance the electron transfer rate [34]. The electrode interface material is extremely important for biofilm establishment in a biofuel cell and affects the resistance of the biofilm against charge transfer. On the other hand, for the long-term monitoring of biorecovery applications, the establishment of biofilms is to be avoided because they affect the electrochemical measurements. In general, highly conductive electrodes are used—for example Pt, Ni, Cu, Au, Fe, Si, CdS, GaAs, glassy carbon, and graphite [25]. The noble metals are chemically inert and the electrodes will not corrode in continuous long-term use. Gold is often selected as one of the most biocompatible metals—for example, in [17,21].

Changes in the electrode potential affect EIS measurements [29], but it has also been found that some bacterial strains will regulate their electron transfer pathways to adapt to the electrode potentials [13]. For microbial activities under anaerobic conditions, an applied CV voltage higher than 0.8 V leads to biological cell rupture, lower growth, and metabolic activity [13].

Unpolished electrodes have a rough surface, resulting in significant differences between the imaginary impedance of most EIS experimental data and capacitor circuit characteristics [29].

### 2.5. Electrical Impedance Spectroscopy

The spectrum of the electrical impedance is determined by measuring both the complex sinusoidal voltage, *V*, and the complex sinusoidal current, *I*, across a range of frequencies. The resulting complex impedance, *Z*, can be determined according the Ohm’s Law relationship
(1)$$Z=\frac{V}{I}.$$

Because all the quantities are complex, they have both a magnitude and a phase. The magnitude of the impedance is given by
(2)$$\left|Z\right|=\frac{\left|V\right|}{\left|I\right|}$$
and the phase is given by
(3)∠Z=∠V−∠I
where |Z| represents the magnitude of *Z* and ∠Z represents the phase of *Z*.

A custom low-cost analyzer designed by the research team [32] achieves the impedance measurement by generating a voltage sinusoid at different frequencies. At each frequency, the voltage is applied across the electrodes and the resultant current is measured simultaneously with the voltage. The magnitude and phase of the two signals are used to calculate the impedance according to Equations (Equation 2) and (Equation 3).

The frequency range to be measured is dependent on the application requirements and can therefore be set up for the specific experiment being done. The device then measures the impedance at each frequency point and the resulting impedance spectrum can be analyzed by means of mathematical impedance modeling. Table 1 shows the electrical setup for the EIS measurements.

### 2.6. Impedance Modeling

Since the suspended ions and bacterial cells in the solution act as reactive impedance elements, the measured impedance data are converted from polar to rectangular form. This allows a polynomial to be fitted to the imaginary part, or reactance, of a complex impedance. A polynomial in the Laplace domain of the form,
(4)$$Z_{fit,s}=a_{0}+b_{0}s^{p_{1}}+c_{0}s^{p_{2}}+d_{0}s^{p_{3}},$$
where s=j×2πf and *f* the frequency at which the measurement took place, is fit to the reactance data. In Equation (Equation 4), $a_{0}$, $b_{0}$, $c_{0}$, and $d_{0}$ are constants used to determine the size of an appropriate electrical component, whereas the exponent values of $p_{1}$, $p_{2}$, and $p_{3}$ inform the type of electrical component. For example, if p=−1, the associated term will be of the form
(5)$$\frac{1}{k_{1}s},$$
which is the Laplace domain definition of a capacitor of magnitude $k_{1}$. If p=1, the associated term takes the form
(6)$$k_{2}s,$$
which corresponds to an inductor of magnitude $k_{2}$. Integer multiples of *p* likely indicate the presence of multiple instances of the same magnitude capacitor or inductor. If there is no frequency dependency, as seen in the term $a_{0}$ in Equation (Equation 4), the term has a real value and is associated with resistance.

An automated curve-fitting function is used in Python to find the optimal fit solution for Equation (Equation 4) to each reactor. This includes an optimizer that checks a maximum of 50,000 iterations to minimize the least squares difference between the data and the proposed model. Once a solution is found for the fit for each of the individual reactors, the average values of the extracted parameters from the three reactors are used for further analysis and discussion. The constants in Equation (Equation 4) are compared to the CFU counts for *K. pneumoniae* and *P. bifermentans* to evaluate whether changes in these constants can be used as a proxy for a change in the observed parameters. A strong correlation would make these constants excellent quantities to use for the inline monitoring of bacterial growth, and, by extension, of the metabolic activity (MA).

## 3. Results and Discussion

### 3.1. Colony-Forming Units

The number of colony-forming units per mL of sample was determined for each bacterial strain in cultures grown with and without an 80 mg/L initial concentration of Pb(II). For further discussions, the term “CFU count” is used to refer to $\log_{10}\left(\mathrm{CFU}/\mathrm{mL}\right)$ unless otherwise specified. The CFU count for the reactors containing *K. pneumoniae* can be seen in Figure 2a and the CFU count for the reactors containing *P. bifermentans* can be seen in Figure 2b.

From Figure 2, it is clear that both strains perform better in the presence of Pb(II), with *K. pneumoniae* reaching a similar maximum CFU count to *P. bifermentans*. It should be noted that the CFU count for *K. pneumoniae* increases rapidly up to approximately 9 h, followed by a slower, but still significant, increase for the duration of the experiment (up to *circa* 22 h). The similarities in the growth trends for *K. pneumoniae* with and without the added Pb(II) could signify that the Pb(II) in the solution markedly increases the growth of *K. pneumoniae*, likely due to the availability of Pb(II) as an anaerobic respiration co-substrate, therefore resulting in increased ATP production [35]. In contrast, the *P. bifermentans* shows an initial increase in CFU count up to a maximum at approximately 9 h, followed by a decrease to the end of the experiment. This is an indication that *P. bifermentans* experiences a substrate depletion at around 9 h, as opposed to sufficient substrate supply for the *K. pneumoniae*. These differences would further imply that any measurements related to growth would be markedly affected by the presence of Pb(II) due, in part, to the differences in the growth profiles in the presence and absence of Pb(II).

The biological and chemical mechanisms employed by bacteria leading to lead resistance are applicable to the bioremediation process and can be utilized in the wastewater industry [5]. These mechanisms include transport across the cell membrane, intracellular bioaccumulation, physical adsorption, surface biosorption, ion exchange, chelation, alteration in cell morphology, precipitation, and the occurrence of a redox reaction. In previous research based on the microbial consortium from which the strains were isolated, it was proposed that an initial surface biosorption occurs, which could possibly occur in the individual strains as an preliminary detoxification mechanism [8].

### 3.2. Residual Pb(II) Measurements

The residual Pb(II) measurements for both *K. pneumoniae* and *P. bifermentans* can be seen in Figure 3. Since initial experiments indicated that *P. bifermentans* exhibited a slower growth rate than *K. pneumoniae* in the presence of Pb(II), the measurements containing *P. bifermentans* were taken later during the runs, thereby giving the strain time to reach exponential growth phase. It is, however, evident that *P. bifermentans* has a faster Pb(II) removal rate over the duration of the experiment. The Pb(II) measurement results provide an explanation for the differences in growth profiles in Figure 2 because it is clear that the Pb(II) concentrations in the *P. bifermentans* reached a minimum after 9 h (corresponding to the maximum CFU count). In contrast, the *K. pneumoniae* Pb(II) removal continued for the duration of the experiments, as reflected in the continued increase in CFU count for *K. pneumoniae*. Both strains reached a similar final value for the Pb(II) concentration.

The changes in Pb(II) concentration can be observed visually as a dark precipitate forming. The visual changes shown in Figure 4 mirror the results of Figure 3.

### 3.3. Electrical Characterization of Bacterial Growth

The Nyquist plot during the growth phase from one bioreactor is generated for samples of *K. pneumoniae* and *P. bifermentans* both with and without lead. The results are shown in Figure 5 and Figure 6 for *K. pneumoniae* and *P. bifermentans* respectively. Similar plots can be generated for all three reactors in a particular measurement set, but a single reactor will suffice to grant initial insights into the most important electrical phenomena during the growth phase. A Nyquist plot shows the resistance of the complex impedance on the *x* axis, and negative reactance on the *y* axis. The negative reactance is used to ensure that the plot is entirely within the first quadrant. Impedance phenomena occurring at low frequencies can be found on the right-hand side of the Nyquist plot, while high-frequency electrical phenomena are found on the left. An inset in each of the plots shows a zoomed-in view of the high-frequency, low-impedance data. It is immediately evident that the Nyquist plots in Figure 5 and Figure 6 do not fit the standard Randles equivalent circuit model well, as was expected.

When comparing Figure 5a,b after 22 h, it is clear that the removal of Pb(II) seen in Figure 3 has significantly increased the resistance and reactance in the system, likely due to the shielding effect of elemental Pb and the decrease in available Pb(II) ions that act as charge carriers. This effect is less pronounced when comparing Figure 5 and Figure 6, which may be due the fact that *P. bifermentans* is a Gram-positive bacterium. Removing positively charged Pb(II) ions through an increase in positively charged bacterial cells seems to obscure the trend that can be seen in Figure 5, suggesting that a more nuanced approach than simply using Nyquist plots is required to study the changing electrical characteristics of these systems. This incomplete understanding in the mechanisms of anaerobic growth of bacteria is a known problem [36], and Equation (Equation 4) is an attempt at eliciting meaning from the measured impedance results.

The hourly EIS data for the exponential growth phase demonstrate what appears to be a semicircle emerging at higher frequency, specifically in the hour 6 and 7 measurements for Figure 5a, and hours 4–7 in Figure 5b. This is characteristic of a resistor in series with a capacitor [24]. Using this insight, it is assumed that $a_{0}$ in Equation (Equation 4) is a constant reactive quantity corresponding to this impedance in Ω and the exponent $p_{1}<0$. The fact that the semicircle is ill-defined indicates that the exponent $p_{1}$ is likely not an integer, which suggests an element that acts similarly to a capacitor with $b_{0}$ its magnitude in Farad. There is a marked downward curve at the extreme high frequencies counteracting the RC response, especially at hour 22 in Figure 5a. This suggests a positive phase shift at high frequencies, likely attributable to inductance in the wires connecting the measurement setup to the measurement device. From this, $p_{2}>0$, with $c_{0}$ the magnitude of an inductor in Henry.

The low-frequency response seen in Figure 5, and particularly in Figure 6, shows an approximately straight line making a 45° angle with the horizontal axis. This corresponds to a Warburg element, which is an impedance element that contributes a constant phase shift across all frequencies [37]. Warburg impedance is associated with the diffusion of charge carriers into the solution, and its magnitude is described by
(7)$$|Z_{W}|=\frac{\sqrt{2}A_{W}}{\sqrt{s}}=\sqrt{2}A_{W}s^{-0.5}.$$

Due to the charge present on the outside of both bacterial cells, as well as ions suspended in the solution due to the LB broth and Pb(II), this is expected electrical behavior. The assumptions that can be made from the Nyquist plot can be used to inform the nature of the polynomial that is fit to each measured dataset. Adjusting Equation (Equation 4) using the above insights leads to
(8)$$Z_{fit,s}=a_{0}+b_{0}s^{p_{1}}+\frac{c_{0}}{\sqrt{s}}+d_{0}s^{p_{2}},\mathrm{with}\;p_{1}<0\;\mathrm{and}\;p_{2}>0.$$

After fitting Equation (Equation 8) on all the hourly data sets gathered for both *K. pneumoniae* and *P. bifermentans*, it was discovered that the powers $p_{1}$ and $p_{2}$ settle very close to the same value in every case. The values for these parameters were then fixed to $p_{1}=-1.5$ and $p_{2}=1$, which correspond to the electrical performance of a fractional power supercapacitor and an inductor, respectively. The former of these may correspond to the proliferation of cells in the solution, while the most likely explanation for the latter is the inductance caused by the wires connecting the sensing electrode to the device used to perform the EIS. The final function used for fitting data is
(9)$$Z_{fit,s}=a_{0}+\frac{b_{0}}{s^{1.5}}+\frac{c_{0}}{\sqrt{s}}+d_{0}s,$$
with the change in $a_{0}$, $b_{0}$, $c_{0}$, and $d_{0}$ over the growing time for *K. pneumoniae* and *P. bifermentans* shown in Figure 7 and Figure 8, respectively.

From Figure 7, it is clear that the large error bars on $-a_{0}$ and the inductance $-d_{0}$ indicate that these parameters cannot reliably be used to estimate the change in CFU count or metabolic activity for *K. pneumoniae*. The scale for $d_{0}$ is 10^−5^, which is too small an inductance to be measured from a bioreactor in situ, and the fact that both $-a_{0}$ and $-d_{0}$ are only affected in the presence of Pb(II) makes them unsuitable for inline measurements of a bioreactor with an unknown chemical makeup. Fixing the value of $-d_{0}$ to 1.5 × 10^−5^ prevented the optimizer from reaching a solution, implying that although the wire inductance is small, it cannot be neglected. Logarithmic scales are required for $-b_{0}$ and $-c_{0}$ because of the large difference in the magnitudes of these parameters with and without Pb(II). This alone is a good indicator that the magnitude of these parameters can be used as a confirmation of the presence of Pb and is also promising for the potential of this method to study the change in Pb(II) concentration. Further, the presence of suspended Pb(II) serves to amplify the change in $-c_{0}$ and $-d_{0}$ during the initial growth phase, supporting the observation that *K. pneumoniae* is well adjusted for the toxic environment, as seen in Figure 2a. From Figure 7, the monotonic change in $-b_{0}$ over the growth period identifies this as the most promising proxy measurement parameter for CFU count.

Similar to *K. pneumoniae*, Figure 8 for *P. bifermentans* shows very little change in $d_{0}$ over the measured growth period, which is suspected because this parameter is related to the measurement setup. Unlike for *K. pneumoniae*, fixing $d_{0}=0$ for *P. bifermentans* still yielded very similar results for the other fit parameters, showing that the polynomial used for the fit is less sensitive to the inductance seen at the highest frequencies considered. The difference between the values seen in $b_{0}$ for reactors with and without Pb(II) shows the potential to extract the Pb(II) concentration using this method, although this difference is much smaller than the comparison seen for *K. pneumoniae*. The slope inversion seen in $-c_{0}$ after hour 9 is in agreement with the inversion seen in the CFU count in Figure 2b at hour 9, demonstrating a definite qualitative agreement between impedance and CFU count for *P. bifermentans*. However, the large error bars seen in $-a_{0}$, $-b_{0}$, and $-c_{0}$ at times before 9 h may be related to the irregular initial growth behavior of *P. bifermentans* in Figure 2b. More data points are required to definitively establish the potential to use this modeling methodology for extracting the growth of *P. bifermentans*. Furthermore, from Figure 3, it can be seen that *P. bifermentans* is much more effective at removing Pb(II) ions during the initial growth phase, which means that, at later sampling intervals, a significant amount of precipitated atomic Pb is still suspended in the solution, affecting the impedance measurements. This is quite different from *K. pneumoniae*, which shows a more consistent rate of Pb(II) removal than *P. bifermentans*, which leads to less suspended Pb precipitate during the growth phase.

Another important aspect that may have a significant impact on the electrical measurements is that *P. bifermentans* is a Gram-positive bacterium, whereas *K. pneumoniae* is Gram-negative. This not only has implications for the metabolic processes of these cells and their interactions with the Pb(II), but also the measured impedance response, due to the different ways that these bacteria interact with charge in the solution. The method in the case of *P. bifermentans* may require the simultaneous quantification of the Pb or, alternatively, a more complex polynomial may be required for fitting the data measured for *P. bifermentans*.

### 3.4. CFU Prediction Using Impedance Measurement

Correlation plots of the impedance fit data against the CFU count data as a time progression for samples with and without Pb(II) are presented in Figure 9 and Figure 10 for *K. pneumoniae* and *P. bifermentans*, respectively. A correlation plot reveals the feasibility of using the EIS measurement method for predicting the CFU count in each case as an indicator of cell growth and metabolic activity. The expected parameter to use for the extraction of cell growth is the supercapacitor corresponding to $-b_{0}$. Supercapacitor chemistry is associated with the interaction of ions in the double-layer capacitor at the interface of an electrolyte and the electrode material, suggesting that a change in the bacterial cell concentration in the solution changes this interaction.

The supercapacitor, $-b_{0}$, curves of *K. pneumoniae* in the presence and absence of Pb(II) against the measured change in CFU count are shown in Figure 9. What is immediately evident is the clear difference over the complete experimental time in the $-b_{0}$ magnitude for an experiment with Pb(II) included, compared to one without. A magnitude threshold can therefore be used to indicate the presence of Pb(II) in a reactor. The shape of the time progression relates to the growth curve of the reactors containing *K. pneumoniae* that can be seen in Figure 2a. During the exponential growth phase in the first two time periods up to 6 h, an approximately straight line prediction of the CFU count from the magnitude of $-b_{0}$ results in both the presence and absence of Pb(II). The monotonically decreasing regression lines indicated on the data in Figure 9 show that this measurement method has great promise as a technique to monitor the cell growth of *K. pneumoniae* in these bioreactor systems. From the growth curves in Figure 2a, it was noted that *K. pneumoniae* performs better in the presence of Pb(II), and this effect is reflected in the extracted regression slopes of the leaded and unleaded cases. For the time periods after 6 h, the shape of the $-b_{0}$ curve is likely significantly affected by the higher levels of suspended elemental Pb related to the lower residual Pb(II) measurements for *K. pneumoniae* as seen in Figure 3. It is yet to be determined what effect vastly different or continually changing Pb(II) concentration would have on the effectiveness of the cell growth measurement, but from the change in Pb(II) concentration seen in Figure 3, the method is robust against significant variations in Pb(II) concentration.

The average time progression correlation behavior of the extracted supercapacitor, $-b_{0}$, for *P. bifermentans* in Figure 10 supports the accordance between the CFU count data of Figure 2b. In the presence of Pb(II), the pronounced inversion on both axes of Figure 10 at the last time point relates to the negative growth rate of *P. bifermentans* after 10 h. A linear relationship is obtained between the supercapacitor value and the CFU count during the exponential growth phase, similar to what is obtained for *K. pneumoniae*.

In the absence of Pb(II) curve of Figure 10, there is also an accordance with the growth curve of Figure 2b. There is a linear relationship in the correlation data for the first and the third time sections, and the irregular vertical line during the second time section corresponds to consecutive measurements with no change in plated CFU count, as seen in the time period of 7 to 8 h in Figure 2b. Although the data are irregular and too sparse to definitively predict the relationship, it can be conjectured that a linear regression is also feasible for the exponential growth phase for *P. bifermentans* in the absence of Pb(II). It is interesting to note that the sign of the slope in this case is different from that of *K. pneumoniae* in the case without Pb(II), which may be due to the fact that *P. bifermentans* is a Gram-positive organism while *K. pneumoniae* is Gram-negative. Different electrical interactions between the cell charge of the Gram-positive organisms and the positive ionic Pb(II) charge are to be expected.

Although focused on the growth of a biofilm on the surface of the electrode, Ref. [26] shows a similar decrease in the impedance response corresponding with an increase in organic biofilm thickness. The decrease in the extracted supercapacitor seen in Figure 9 and Figure 10 is in good agreement with similar EIS-based measurements aimed at classifying bacterial growth [27].

### 3.5. Continuous Monitoring Devices

The low-cost device used for the EIS measurement in this investigation can be reproduced for implementation on commercial bioreactors. Data collection on this device occurs via USB to a computer interface. If this device were to be implemented in a continuous lead recovery bioreactor, a more robust and possibly remote data transfer method would be required, as well as a power supply that does not interfere with the functioning of the bioreactor. The electrodes used for this investigation are gold patterned on a plastic substrate, and are made to be discarded. For continuous measurement in a bioreactor system, custom low-maintenance electrochemical electrodes with a long lifespan must be designed. To manufacture these electrodes, the use of printed electronics such as screen printing or inkjet printing may be a significant cost-saving measure. Similarly, the use of additive manufacturing methods may be used to manufacture enclosures and fittings to facilitate ideal device and electrode placement, similar to the jig used in this work, as seen in Figure 1. Going further, custom application-specific integrated circuits have been shown to hold potential for point-of-need diagnostic testing. If such a device is implemented with hybrid integration of bare die packaging, the integrated circuit can include electrodes with feature sizes of the same orders of magnitude as the bacterium under investigation. Any other improvements to such a device, including extending its frequency measuring range or further data processing, may be included on one integrated circuit.

## 4. Conclusions

The growth and Pb(II) concentration data presented in the study provide strong evidence of the effect of Pb(II) as an anaerobic respiration co-substrate for both *K. pneumoniae* and *P. bifermentans*, resulting in the production of elemental Pb. These observations provide strong support for the potential of this process to biovalorize Pb(II) as elemental Pb. It is of particular interest how the changes in the CFU count measurements were qualitatively mirrored in the changes in the Pb(II) concentrations. At the apparent depletion of Pb(II) in solution, the *P. bifermentans* reached a maximum CFU followed by a growth decrease, while the consistent presence of Pb(II) in solution due to a slower depletion rate resulted in the consistent growth of *K. pneumoniae*. These trends were further qualitatively mirrored in the observed impedance measurements, therefore providing strong support for the potential for the inline, continuous, non-invasive monitoring of the growth using these measurement techniques.

Clear trends can be observed that give a direct relationship between the supercapacitor magnitude and CFU during the exponential growth phase for both *K. pneumoniae* and *P. bifermentans* in the presence of Pb(II). Due to the effect that suspended Pb(II) has on the absolute impedance magnitudes, it is advised that the relative difference between samples as time progresses is used as an indicator for cell growth. Once the exponential phase has passed, only qualitative trends can be extracted. A differentiation between Gram-positive and Gram-negative bacteria can be made using the trends seen in any of the four circuit parameters used for data fitting, as the two different bacteria show opposing trends. This or a similar investigation would benefit greatly from a larger dataset of measurements taken over a longer time to fully classify the relation between the impedance and the cell growth and the Pb(II) concentration.

Impedance spectroscopy offers a suitable means for the qualitative monitoring of the metabolic activity of bacteria in a lead recovery environment. The quantity extracted from the measured data correlates with a composite of the metabolic substrate utilization, cell respiration and mediator excretion, and biomass production and decay. The complexity causes a challenging lack of mechanistic insight [36]. For quantitative monitoring, further investigation of the interface reactions is required, including biotic and abiotic processes on the electrode surface. Expansion of the impedimetric technique may reveal the electron transfer mechanism between a microorganism and the suspended lead and can enable the analysis of the physiological and metabolic characteristics of the different cells in a mixed microbial community. In future research, improved specificity can be achieved by either extending the bandwidth of the measurement or by utilizing different electrodes. The frequency range can either be extended lower to enable diffusion-related measurements or it can be extended higher to more precisely examine high-frequency dielectric parameters where the presence of cells would have a more significant contribution. Similarly, the impedance contributions of different phenomena can be manipulated using different electrodes. Multiple and different electrodes could be used to quantitatively extract more system parameters. Optimizing the construction of all the system components (i.e., electrodes, cell design) will contribute to the improved performance of the process. Better understanding of the bioreactor ecology, along with the monitoring of multiple parameters related to the microbe, the substrate utilization, and the media, will provide rational control capabilities to the system.

In a permanent installation, electrode biofouling is a challenge that may require sufficient maintenance, with the associated time and cost [31]. Here, an appropriate sampling time is proposed after three cleaning cycles of CV at a 2 V amplitude and a scan rate of 1 V/s. Cleaning CV cycles have demonstrated the restoration of fouled electrodes for real-time remote monitoring over the longer term [26], and the efficiency of this method must be evaluated in the lead biorecovery application using *K. pneumoniae* and *P. bifermentans*.

The microbial community and the measurements in the complex system may be affected by the electrolyte conductivity and the pH of the medium, as well as ambient temperature, external electromagnetic fields, and light irradiance, and investigations are required regarding these effects.

In summary, EIS promises the discrimination and quantification of key factors limiting the performance to support the commercial scaling up of lead biorecovery. Experimental bioreactor integration with the custom low-cost impedance device opens substantial opportunities for future scaling via efficient inline monitoring and automated control of the bioreactor process, which is crucial to large-scale continuous lead recovery processes.

## Figures and Tables

**Figure 1 sensors-22-07045-f001:**
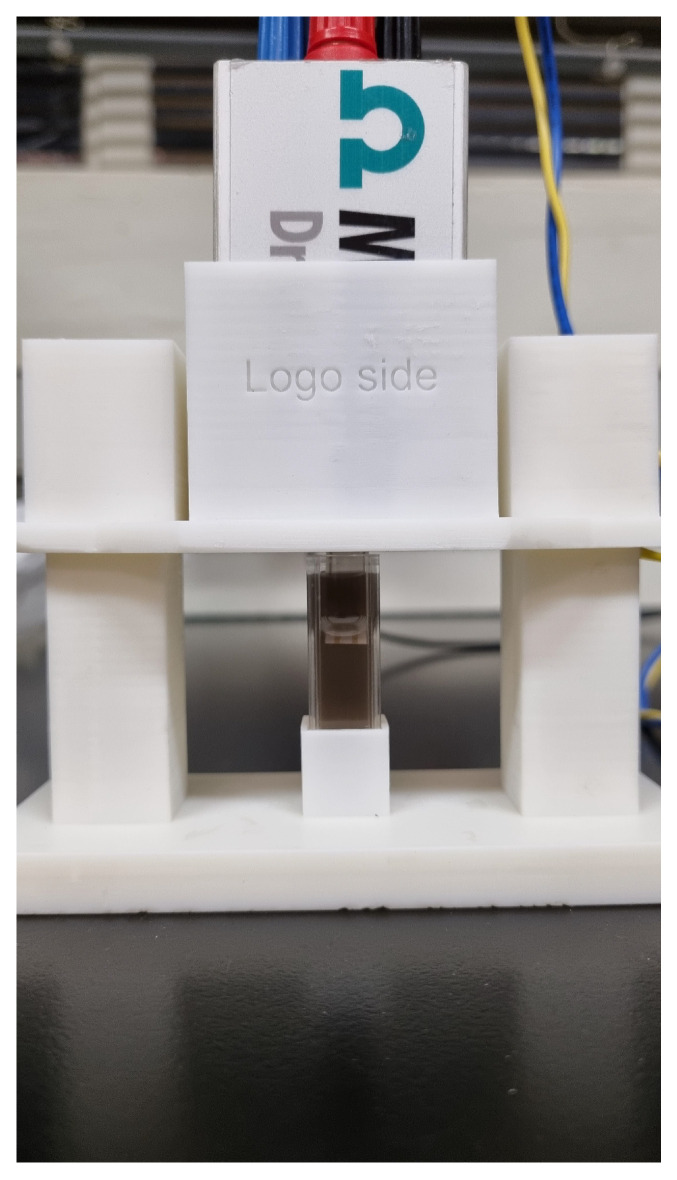
Custom 3D-printed jig for electrical measurements, showing the electrode connector keeping an electrode in suspension in a cuvette with a 1.8 mL sample.

**Figure 2 sensors-22-07045-f002:**
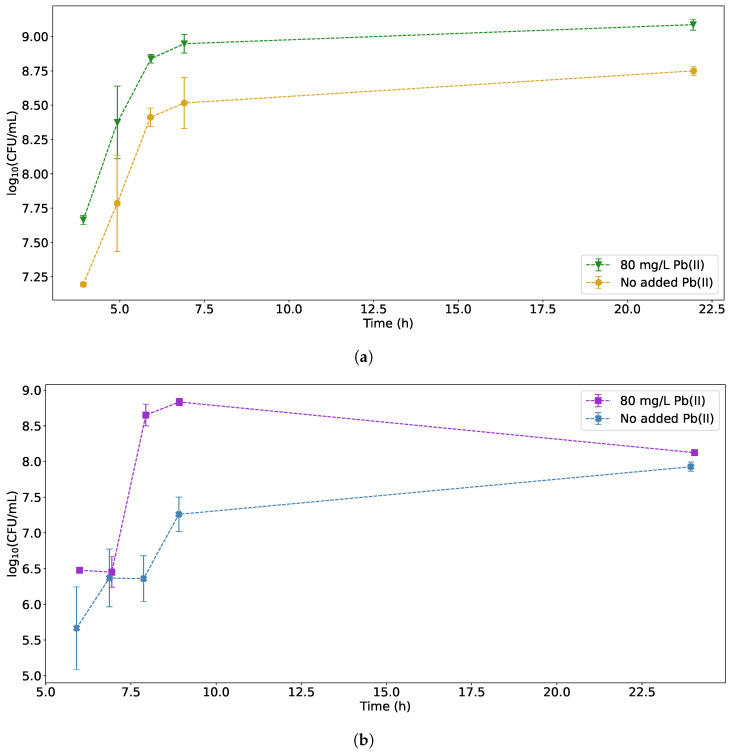
Growth curves for (**a**) *K. pneumoniae* and (**b**) *P. bifermentans* with and without addition of Pb(II) to the solution.

**Figure 3 sensors-22-07045-f003:**
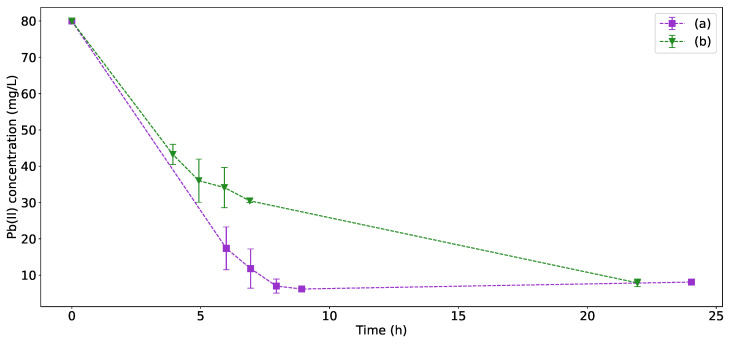
Residual Pb(II) concentration in reactors containing 80 mg/L initial concentration of Pb(II) with (**a**) *P. bifermentans* and (**b**) *K. pneumoniae*.

**Figure 4 sensors-22-07045-f004:**
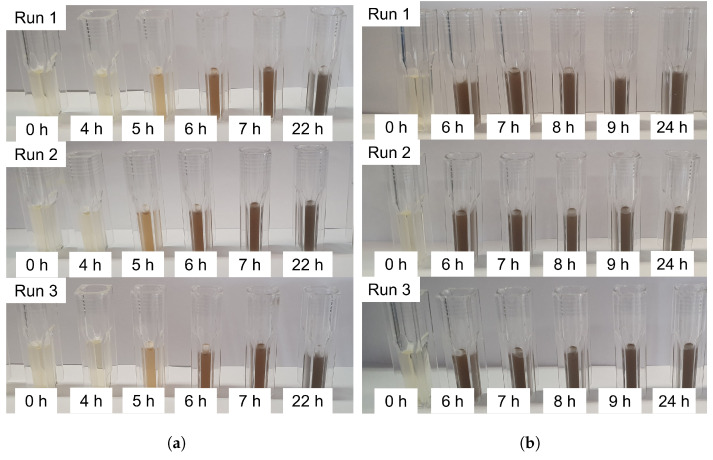
The visual results of the three duplicates for both experiments containing 80 mg/L initial Pb(II) concentration. (**a**) The visual results for the experiment involving *K. pneumoniae*. (**b**) The visual results for the experiment involving *P. bifermentans*.

**Figure 5 sensors-22-07045-f005:**
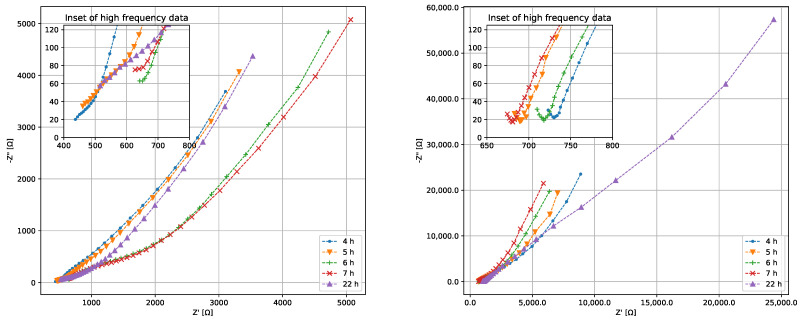
Nyquist plots of the EIS data for *K. pneumoniae* (**a**) without and (**b**) with Pb(II).

**Figure 6 sensors-22-07045-f006:**
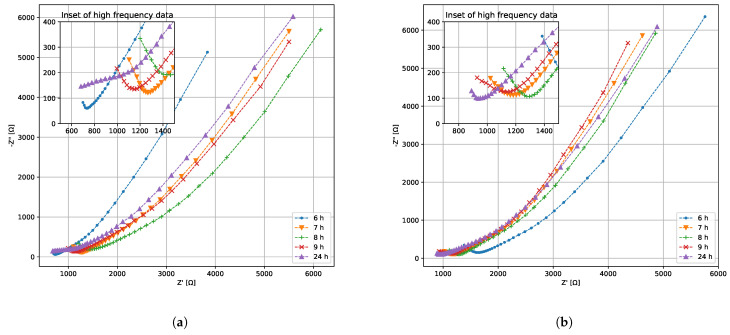
Nyquist plots of the EIS data for *P. bifermentans* (**a**) without and (**b**) with Pb(II).

**Figure 7 sensors-22-07045-f007:**
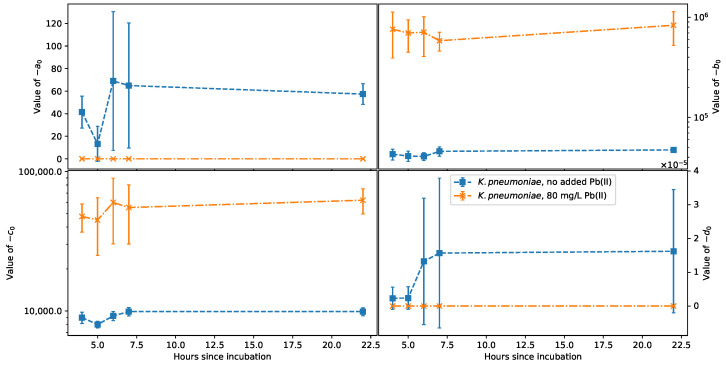
Change over time of the fit parameters $a_{0}$, $b_{0}$, $c_{0}$, and $d_{0}$ for *K. pneumoniae*.

**Figure 8 sensors-22-07045-f008:**
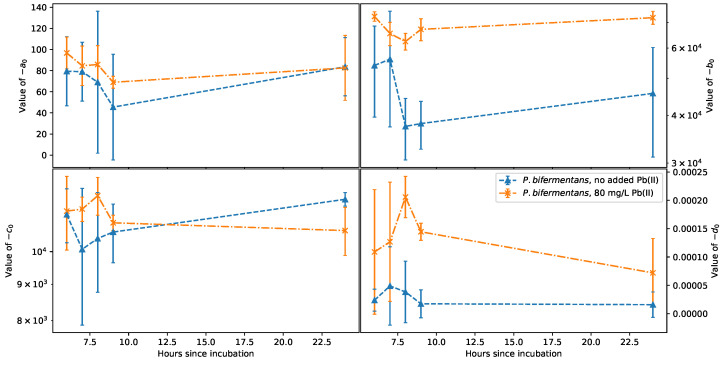
Change over time of the fit parameters $a_{0}$, $b_{0}$, $c_{0}$, and $d_{0}$ for *P. bifermentans*.

**Figure 9 sensors-22-07045-f009:**
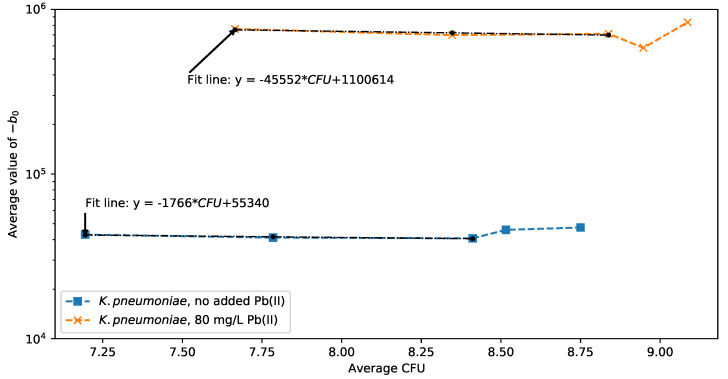
Changes in the supercapacitor $b_{0}$ for *K. pneumoniae* with and without Pb(II).

**Figure 10 sensors-22-07045-f010:**
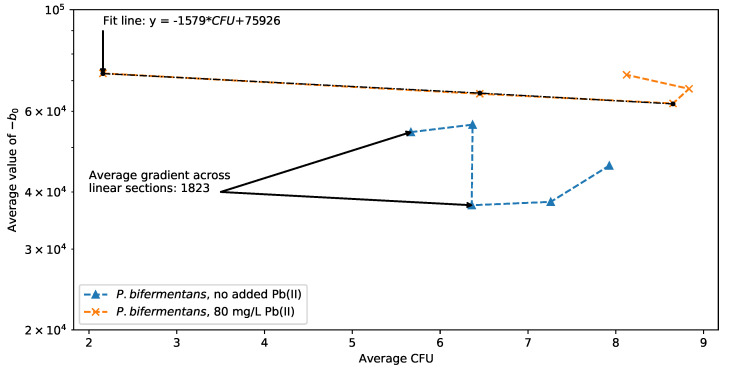
Changes in the supercapacitor $b_{0}$ for *P. bifermentans* with and without Pb(II).

**Table 1 sensors-22-07045-t001:** Settings used for EIS experiment.

Parameter	Value	Unit
Minimum frequency	10	Hz
Maximum frequency	1	MHz
Excitation amplitude	1	V
Minimum measured current	150	μA
Maximum measured current	100	mA

## Data Availability

The data used in this study are publicly available in Mendeley Data at https://doi.org/10.17632/jfvzvwzggv.1.

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
