# Peer review of "Non-Destructive Impedance Monitoring of Bacterial Metabolic Activity towards Continuous Lead Biorecovery"

_sensors, 2022, doi:10.3390/s22187045_

Round 1
Reviewer 1 Report
Comments: Overall, the manuscript is well written. Some of the comments to be addressed are:
- The introduction part is to be rewritten, highlighting the other methods proposed for selecting microorganisms’ consortia with high metabolic activity.
- Insert a figure showing a redox-reactive biosensor for extracellular bacterial metabolic analysis as supplementary material.
- Compare the results obtained in Figs. 5 and 6. Why was this behavior change observed? Explain.
- Check and correct the typographical and grammatical mistakes throughout the manuscript.
Author Response
The research was focused on using bacterial strains which have been isolated from an industrially obtained lead-resistant consortium. The focus is therefore placed on the lead resistance of the strains and not the high metabolic activity. A sentence has been added to the introduction to make this clearer.
A section was added to the introduction that discusses a number of alternative bacterial metabolic activity monitoring methods and why the method used is most appropriate for this investigation. Cyclic voltammetry for redox reactive electrochemical setups is among the methods discussed.
The results shown in Figs. 5 and 6 are used to guide the development of a model that is used for a more detailed comparison in later sections. A brief comparison and this clarification have been added.
Typographic and grammatical errors have been addressed.
Reviewer 2 Report
The work entitled “Non-destructive impedance monitoring of bacterial metabolic activity towards continuous lead biorecovery”, authors are George Andrews, Olga Neveling, Dirk Johannes de Beer, Evans M.N. Chirwa, Hendrik G. Brink, and Trudi-Heleen Joubert, is devoted to a very important problem of contamination of the environment with heavy metals having strong toxic effects for ecosystems and living beings. Authors presented the results of experiments for detection of microorganisms able to transform Pb ions into the reduced metal form. This anaerobic biological process provided detoxification of Pb(II) and its accumulation for further possible applications and regeneration of lead as a mineral source. The work is original and novel. A rare method of electrical impedance spectroscopy (EIS) for estimation of metabolic activity and number of Pb accumulating bacteria is discussed in details.
Specific comments
1) Abstract: It is recommended to give genera names of microorganisms in full.
2) Materials and Methods: Measurement units are lost for all values in the section Materials and Methods, and this loss of units must be carefully checked in other sections too.
3) Materials and Methods: It is recommended to indicate what Pb(II) salt was used and indicate the manufacturer of the Pb(II) solution / Pb (II) salt.
4) Figures 2, 3 – It is better to transfer descriptions for variants a and b to the figure caption, do not give them directly on the picture.
5) Usage of Pb2+ as anaerobic co-substrate is appropriate explanation for good growth of the studied microorganisms. However, Pb(II) is a known toxic agent. To improve this paper, discussion about possible mechanisms of K. pneumoniae and P. bifermentans resistance against Pb(II) and why these toxic effects are not observed in experiments could be added into the text.
6) Figure 3, caption – “concentration of reactors” should be changed with “concentration in reactors”.
7) Figures 5, 6 – It is recommended to not describe a and b directly in the figures but add a and b in the caption text: “Nyquist plots of the EIS data for K. pneumoniae (P. bifermentans) without (a) and with (b) 80m? Pb(II)”. It is recommended to write time as
4 h 6 h
5 h 7 h
6 h 8 h
7 h 9 h
22 h 24 h
Summary
Recommend for publication after minor revision

Author Response
1) Genera names of microogranisms given in full in the Abstract
2) Measurement units for the entire manuscript disappeared between submission and distribution to reviewers. This matter has been brought to the attention of editors.
3) The Pb(II) salt was indicated and a method of preparation of the Pb(II) solution was added to the materials and methods section.
4) Altered.
5) Added to the discussion under “colony forming units”.
6) Altered.
7) Altered.